# Spatial tuning and brain state account for dorsal hippocampal CA1 activity in a non-spatial learning task

Kevin Q Shan, Evgueniy V Lubenov, Maria Papadopoulou, Athanassios G Siapas*

Division of Biology and Biological Engineering, California Institute of Technology, Pasadena, United States

**Abstract** The hippocampus is a brain area crucial for episodic memory in humans. In contrast, studies in rodents have highlighted its role in spatial learning, supported by the discovery of place cells. Efforts to reconcile these views have found neurons in the rodent hippocampus that respond to non-spatial events but have not unequivocally dissociated the spatial and non-spatial influences on these cells. To disentangle these influences, we trained freely moving rats in trace eyeblink conditioning, a hippocampally dependent task in which the animal learns to blink in response to a tone. We show that dorsal CA1 pyramidal neurons are all place cells, and do not respond to the tone when the animal is moving. When the animal is inactive, the apparent tone-evoked responses reflect an arousal-mediated resumption of place-specific firing. These results suggest that one of the main output stages of the hippocampus transmits only spatial information, even in this non-spatial task.

## Introduction

*For correspondence: thanos@ caltech.edu

Competing interests: The authors declare that no competing interests exist.

The hippocampus is a brain structure that is known to play a critical role in memory formation (*Squire, 1992*), but the scope and nature of hippocampal memory processing remains elusive. The discovery of place cells in rodents led to the hypothesis that the hippocampus forms a 'cognitive map' that is essential for spatial learning (*O'Keefe and Nadel, 1978*). In contrast, human studies have indicated a more general role of the hippocampus in the formation of episodic memories (*Scoville and Milner, 1957*). In support of the latter view, electrophysiological studies in rodents have found that pyramidal neurons in the dorsal CA1 area of the hippocampus can change their firing in response to arbitrary stimuli, both spatial and non-spatial (*Berger et al., 1976*; *Wood et al., 1999*; *Moita et al., 2003*; *Komorowski et al., 2009*). However, the functional significance of these observations remains controversial because many non-spatial tasks are studied in restrained animals or under circumstances where spatial and non-spatial influences on hippocampal firing cannot be unequivocally dissociated (*O'Keefe, 1999*).

To overcome this challenge, we selected a simple learning task—trace eyeblink conditioning— that can be performed independently of spatial location. In this task, a neutral conditioned stimulus (CS, a tone) is followed, after a short delay, by an unconditioned stimulus (US) that evokes an involuntary blink (*Figure 1A*). With repeated presentations, the subject learns that the CS predicts the US and begins to blink in anticipation (*Figure 1C–D*). Learning the CS-US association has been shown to be hippocampally dependent in many species, including humans and rodents (*Clark and Squire, 1998*; *Kim et al., 1995*; *Takehara et al., 2003*). Previous electrophysiological studies have found that dorsal CA1 pyramidal neurons change their firing following the CS onset (*Berger et al., 1976*, *1983*; *Weiss et al., 1996*; *Weible et al., 2006*), suggesting that these cells encode the non-spatial

**Figure 1.** CA1 pyramidal responses during trace eyeblink conditioning in a freely moving rat. (**A**) An eyeblink trial is a sequence of a tone (CS), a stimulus-free period (trace interval), and a blink-inducing electrical pulse (US). Blinks are measured on an eyelid electromyogram (EMG). (**B**) Trials are delivered randomly throughout the environment as the rat traverses a linear track for water reward. (**C**) Eyelid EMG from early and late in learning. A conditioned response (CR) is defined as an increase in EMG power anticipating the US onset. (**D**) Learning is apparent as an increase in CR frequency over the course of training. (**E**) Example pyramidal cell that significantly increased its firing rate following the CS onset. If spike counts on each trial were Poisson distributed according to this cell's average response (top), we would expect to observe this CS-evoked increase in 76% of trials. Instead, its spike rasters (bottom) show that the response is much less reliable and occurs in only 32% of trials. (**F**) Observed vs. expected CS response reliability. All 1264 recorded pyramidal cells (back dots) responded in less than half of trials. The example from (**E**) is circled in red.

The following figure supplements are available for figure 1:

**Figure supplement 1.** Location of recording sites.

**Figure supplement 2.** Distribution of animal location and velocity during eyeblink trials.

stimulus. However, these experiments did not characterize the spatial tuning properties of the recorded neurons.

Here, we disentangle the spatial and non-spatial influences on hippocampal firing by studying trace eyeblink conditioning in freely moving rats. Consistent with previous studies, we find that the CS can evoke significant changes in the firing rate of dorsal CA1 neurons. Additionally, we find that these apparent CS responses are spatially modulated, but surprisingly, they are absent when the animal is moving. These CS-evoked changes occur only when the animal is inactive, produce firing rates that match normal place cell activity, and are accompanied by a cessation of ripples. These observations are more consistent with an arousal-mediated resumption of place cell firing rather than a genuine encoding of the non-spatial stimuli. These results suggest that neurons in dorsal CA1, a major output stage of the hippocampus, transmit exclusively spatial signals even in this hippocampally dependent non-spatial task.

## Results

To simultaneously map the spatial and non-spatial influences on hippocampal firing, we trained adult rats in trace eyeblink conditioning while they traversed a linear track (*Figure 1B*), an environment in which the spatial coding of hippocampal neurons has been well-characterized (*O'Keefe, 2007*). We presented the eyeblink trials at randomized locations throughout the environment, which enables us to dissociate spatial and non-spatial influences on neuronal activity.

Using chronically implanted tetrode arrays, we recorded from pyramidal cells in the dorsal CA1 region of the hippocampus (an average of 36 cells per training session for a total of 1264). Consistent with previous studies, we found that a subset of cells significantly changed their firing rate following the CS onset (p<0.01 for 10% of cells, two-sided Wilcoxon signed-rank test). However, an analysis of single-trial responses reveals that these cell-level statistics belie a high trial-to-trial variability, and none of the recorded cells respond consistently to the CS onset (*Figure 1E–F*).

What is the source of this trial-to-trial variability? Does spatial tuning play a role? To address this question, we mapped each cell's spatial tuning by measuring its average firing rate as a function of the animal's location while moving around the environment. Cells that were active in the environment had directional place fields distributed throughout the linear track (*Figure 2A*). The remaining cells had low firing rates everywhere, consistent with place cells that lack a place field in this environment.

These spatial firing rate maps—which were constructed with eyeblink trials omitted—reveal that a cell's spatial tuning exerts a strong influence on its response to non-spatial stimuli. For example, the intermittently CS-responsive neuron shown in *Figure 1E* had a place field in the right endbox, and the trials in which it fired corresponded to the trials presented within this place field (*Figure 2B*). To visualize the influence of this spatial tuning, we ordered the trials by the place field intensity at the trial delivery location (*Figure 2B*, bottom panel). Overall, we find that the CS evokes an increase in firing when delivered within a cell's place field, and silences the cell when delivered outside its place field (*Figure 2C–E*). This correlation between CS-evoked firing and place field intensity also holds at the single-cell level (*Figure 2F*). These observations indicate that there are no CA1 neurons that respond consistently to the CS in a space-invariant fashion. Instead, the non-spatial stimulus modulates the firing of place cells.

What is the nature of this modulation? If the CS amplifies ongoing place cell activity, as *Figure 2E* suggests, then we would expect that CS responses would be most prominent when place cells are actively engaged, such as during running. Surprisingly, we find the opposite: CS-evoked changes in firing rate are absent when the animal is running (*Figure 3A*) and occur only when the animal is inactive (*Figure 3B*). Importantly, the lack of CS-evoked changes while running is not accompanied by an impairment of task performance (*Figure 3—figure supplement 1*). When the animal is sitting, the CS-evoked change in firing actually reflects the fact that the pre-trial firing rates differ from the spatially predicted rates (*Figure 3C*, grey markers). After the CS onset, firing rates re-align with the place field intensity at the present location (*Figure 3C*, blue markers). The US evokes a response that briefly exceeds the spatially predicted rates (*Figure 3C*, red markers), but analysis of this phenomenon is potentially confounded by head movement associated with the unconditioned response.

Why do pre-trial firing rates in sitting animals differ from the place field intensity? During periods of inactivity, animals may enter a state of quiet wakefulness, in which the firing of place cells is not exclusively governed by the animal's present location. Instead, the hippocampus exhibits other patterns of firing, such as synchronous bursts (ripples) that may engage neurons with distant place fields (*Foster and Wilson, 2006*; *Csicsvari et al., 2007*; *Diba and Buzsáki, 2007*). We find that when the animal is sitting, ripples spontaneously occur during the pre-trial period, and abruptly cease following the CS onset (*Figure 4A*). Furthermore, CS-evoked changes in firing rate are more pronounced in trials with detected ripples (*Figure 4B*) and the firing of these cells outside their place fields is closely associated with the ripple events (*Figure 4C*). These observations suggest that the animal is indeed in a state of quiet wakefulness during 'sitting' trials, and that the CS triggers an arousal response that leads to a resumption of place cell firing.

Finally, we investigated the effect of the CS on hippocampal theta oscillations. Previous studies have reported that the CS can reset the theta rhythm, aligning the phase of theta to the CS onset (*Moita et al., 2003*; *Nokia et al., 2010*; *Darling et al., 2011*). To distinguish a genuine phase reset

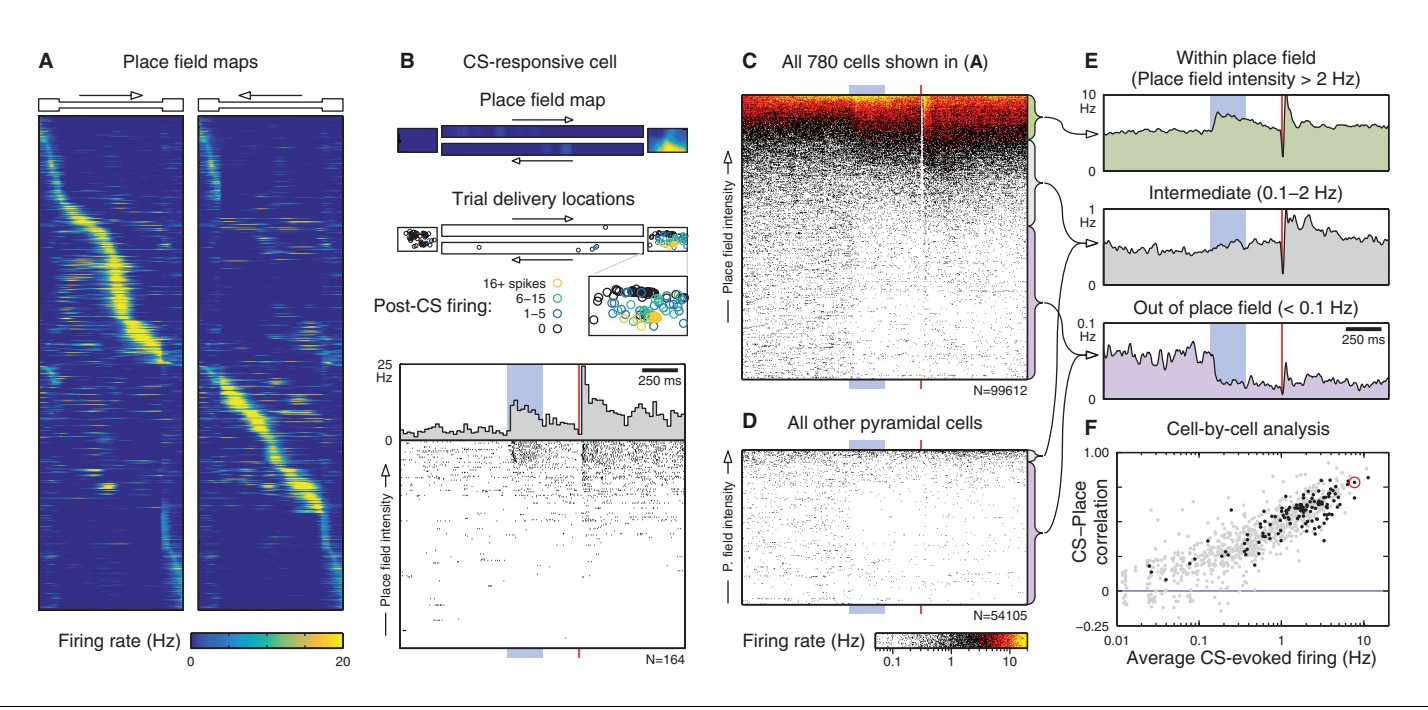

**Figure 2.** Spatial tuning modulates CA1 responses to eyeblink stimuli. (**A**) Spatial firing rate maps (place fields) for the 780 cells active on the linear track. Each row is a single cell; left and right panels show different directions of traversal. These maps are computed excluding eyeblink trials. (**B**) Place fields predict CS responses for the example cell from *Figure 1E*. Top: Place field map. Boxes are the track endboxes, and the track itself is duplicated to show different directions of traversal. Middle: Animal location during eyeblink trials, color-coded by CS-evoked firing. Bottom: Spike rasters ordered by place field intensity at trial delivery locations. (**C**) Spike rasters for all cells in (**A**), ordered by place field intensity. For visualization, many rasters are compressed onto a single row, and overlapping spikes are given warmer colors. Electrical artifacts prevent spike detection during the 10 ms US delivery window. (**D**) Spike rasters for the remaining 484 cells not active on the linear track. (**E**) Average firing rate over all cells and trials, grouped by place field intensity. (**F**) Cell-by-cell analysis of the correlation (Spearman's $\rho$) between CS-evoked firing rate and place field intensity. Most cells have a large positive correlation, especially those that appeared significantly CS-responsive (black dots). The only cells without a positive correlation are those that fired very few spikes. The cell from (**B**) is circled in red.

The following figure supplements are available for figure 2:

**Figure supplement 1.** Additional simultaneously-recorded cells.
**Figure supplement 2.** Additional single-cell examples.
**Figure supplement 3.** Spatial tuning predicts CS- and US-evoked responses on a single-cell basis.

from stimulus-evoked initiation of theta oscillations, it is important to restrict our analysis to trials with ongoing theta activity prior to the CS onset. The prominent theta oscillations during running provide the ideal conditions for this analysis (*Figure 5A*). Under these conditions, we found that both the CS and US can evoke a sustained alignment of theta phase, but only in one out of three animals (*Figure 5D*). Even when theta phase alignment did occur, the new phase remained highly correlated with the pre-stimulus phase (*Figure 5E*), indicating that the observed alignment is due to a subtle phase shift rather than a reset of the theta rhythm. These results confirm that it is possible for the eyeblink stimuli to perturb ongoing theta oscillations, but the absence of this perturbation in two out of three animals shows that it is not necessary to perform the task.

## Discussion

These findings require a re-evaluation of how dorsal CA1 pyramidal neurons may encode non-spatial information. The simplest form of non-spatial coding would be a purely non-spatial *CS cell*

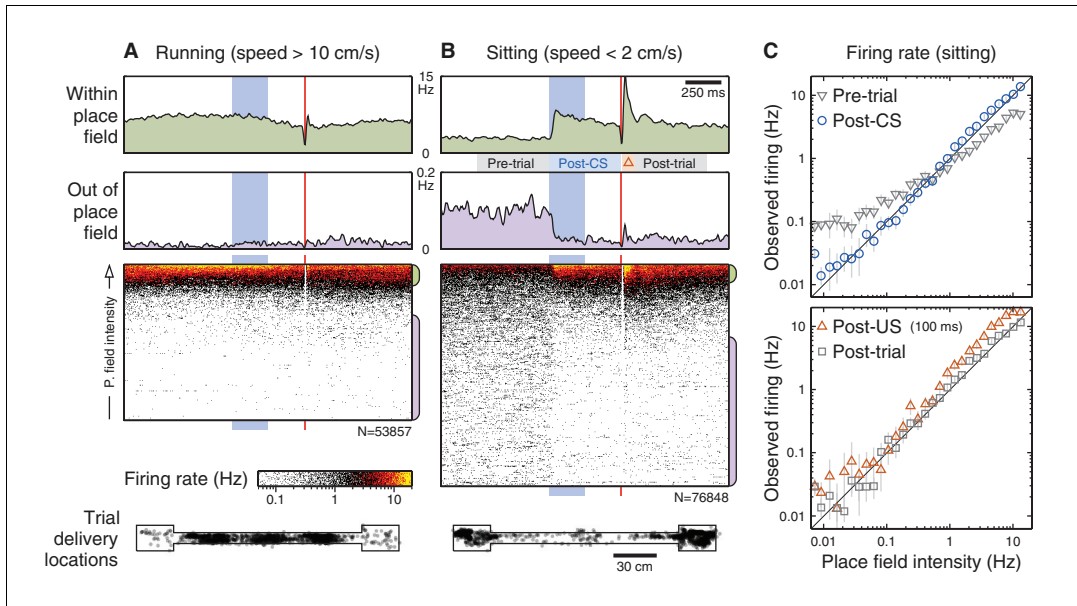

**Figure 3.** Brain state modulates CA1 responses to eyeblink stimuli. (**A**) Responses on trials delivered while the animal was running (40% of all trials). Top: Average firing rate among trials presented within (green) and out of (purple) the place field (same thresholds as *Figure 2E*). Middle: Spike rasters ordered by place field intensity. Bottom: Distribution of animal locations for this set of trials. (**B**) Responses on trials delivered while the animal was sitting (46% of all trials). (**C**) Top: Firing rates, on sitting trials, immediately before (grey) and after (blue) the CS onset. Note that the post-CS firing matches the place field intensity more closely than the pre-CS firing does. Each marker shows the mean and bootstrapped 95% confidence interval for a small range of intensities; values less than 0.01 Hz are not shown to scale. Bottom: Same analysis showing the predicted vs. observed firing rates immediately after the US (red) and shortly after that (grey). The time windows for analysis are indicated in panel (**B**).

The following figure supplements are available for figure 3:

**Figure supplement 1.** Eyeblink performance is not impaired in running trials.

**Figure supplement 2.** State-dependent responses, separated by location.

(*Figure 6A*) that exhibits a consistent change in firing to the CS independent of the animal's location. None of the 1264 pyramidal cells responded in this purely non-spatial manner.

Previous studies that found spatial correlates in hippocampal responses to non-spatial stimuli have hypothesized that these cells encode the conjunction of a specific non-spatial stimulus with a spatial context (*Wood et al., 1999*; *Moita et al., 2003*). A conjunction *CS-place cell* (*Figure 6B*) would change its firing in a location-dependent manner. At first glance, our data appear to fit this model—the CS responses of place cells are gated by their place fields (*Figure 2*)—but our key observation is that CS-evoked changes in firing rate disappear when the animal is running (*Figure 3*). This new finding rejects the conjunctive model because these cells can no longer encode the CS-place conjunction in this behavioral state.

A model that only relies on the known properties of place cell firing provides a more compelling explanation of the experimental observations (*Figure 6C*). In particular, place cells exhibit place-specific firing only when the animal is moving or alert, and exhibit a different pattern of firing when the animal is in a state of quiet wakefulness. If the CS were able to trigger an arousal response, this would manifest as an increase in firing within a cell's place field (due to activation of place-specific firing) and a decrease in firing outside the place field (due to cessation of ripple-related firing). Consistent with this hypothesis, we find that in sitting animals, the CS indeed activates place-specific firing, silences out-of-field firing, and abolishes ripples. When the animal is already alert, such as during running, we observe no additional modulation beyond normal place field firing.

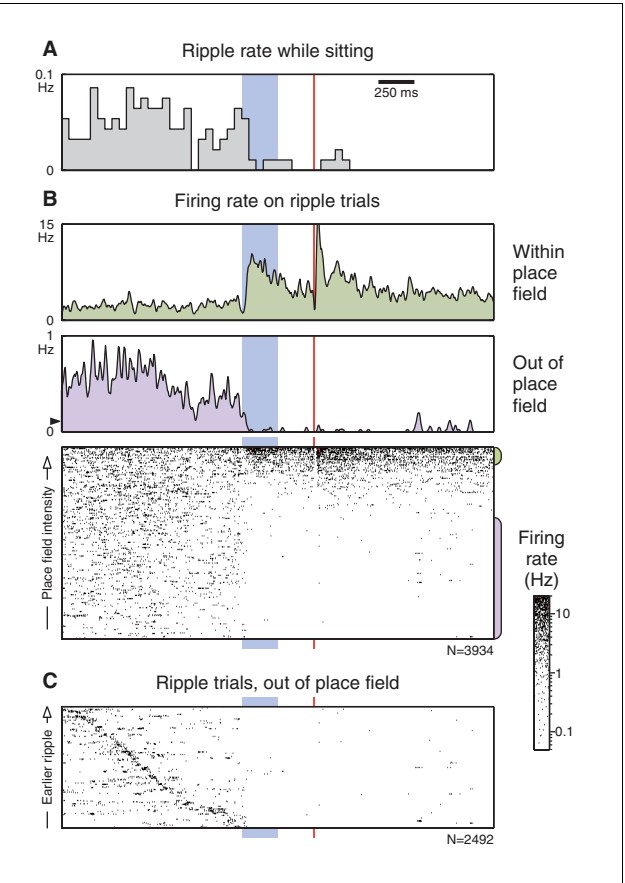

**Figure 4.** Ripples cease following the CS onset. (A) Ripple occurrence rate during trials presented while the animal was sitting (ripples did not occur when the animal was running). (B) Unit responses on trials where a ripple was detected prior to the CS onset. Layout is similar to *Figure 3B*, but note the difference in scale for the 'out of place field' peri-event firing rate. The black triangle marks the average pre-trial firing rate from the corresponding panel in *Figure 3B*. (C) 'Out of place field' rasters sorted by the time that the ripple occurs. Some trials had more than one ripple; for these the longest ripple was used.

Hence the most parsimonious explanation of the experimental data is that dorsal CA1 neurons are all place cells and the apparent responses to the non-spatial stimuli are due to an arousal-mediated resumption of place-specific firing.

These observations clarify the nature of previously-reported patterns of hippocampal activity in eyeblink conditioning. Studies of inactive subjects in a fixed location, such as those with restrained animals (*Berger et al., 1983*; *Weiss et al., 1996*; *Weible et al., 2006*), would find that a subset of cells—those with a place field overlapping the experiment location—consistently increase their firing following the stimulus presentation. Most cells would not have place fields overlapping this location and would be silenced following the stimulus presentation.

The arousal-mediated nature of the hippocampal response can also account for a variety of learning-related phenomena. In particular, any salient stimulus should be able to evoke these hippocampal responses, regardless of whether the task is hippocampally dependent (*Berger et al., 1983*; *Moita et al., 2003*; *Abe et al., 2014*). The salience of the stimulus, and hence its ability to trigger alertness, may change as the subject learns the predictive power of the stimulus, producing a stereotyped evolution of hippocampal activity across learning (*McEchron and Disterhoft, 1997*). On the other hand, manipulations that decrease the salience of the CS, such as habituation to the unpaired stimuli presented to control animals, would reduce the apparent response (*Berger et al., 1976*; *Weiss et al., 1996*).

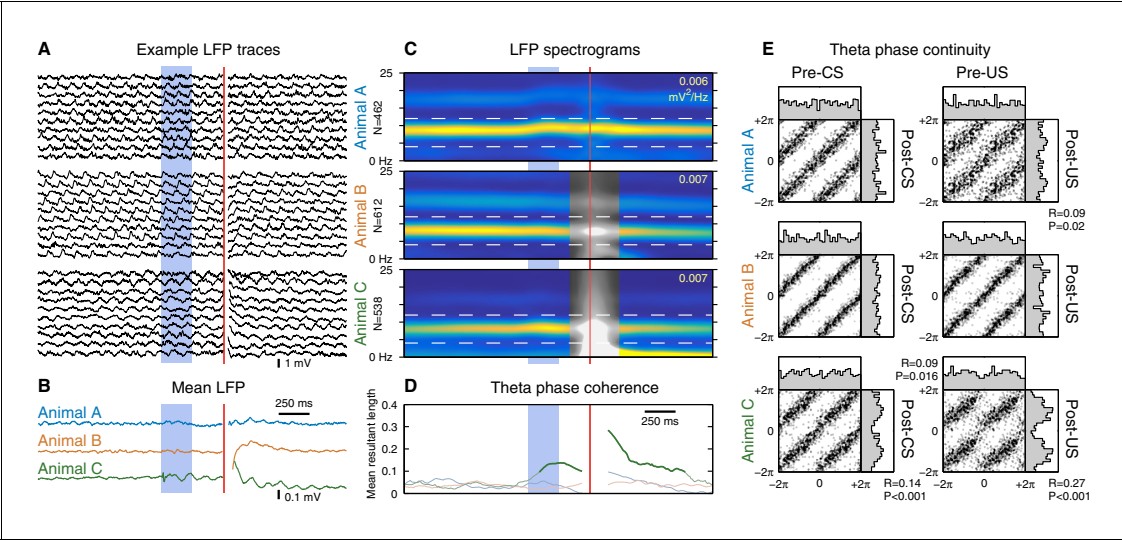

**Figure 5.** The CS does not reset ongoing hippocampal theta oscillations. This figure shows CS- and US-evoked changes to ongoing theta oscillations in the hippocampal local field potential (LFP) while running. (A) Randomly selected example LFP traces from each animal. Note that theta oscillations are prominent and consistent across trials. (B) Averaged hippocampal LFP. Theta-band oscillations are visible in animal C, but note the 10-fold reduction in scale compared to the raw LFPs. (C) LFP spectrograms. Note the strong theta power that persists throughout the trial. Horizontal dashed lines indicate the 4–12 Hz theta band, and the peak spectral power density is indicated in the top right of each spectrogram. Animals B and C exhibited high-amplitude transients following the US, and time windows that overlap this transient are shown in greyscale. (D) Mean resultant length, a measure of theta phase coherence across trials. Values that correspond to a significantly non-uniform phase distribution (p<0.01, Rayleigh test) are shown in bold. Animal C exhibits a sustained phase alignment that persists for several theta cycles, but the other two animals show no significant phase alignment at all. (E) Phase transition curves. Each scatterplot compares the theta phase before and after the stimulus (see Materials and methods and *Figure 5—figure supplement 3*); each dot is a single trial and two cycles are shown for clarity. A diagonal line (e.g. animal B) indicates that the theta rhythm is unaffected by the eyeblink stimuli, whereas a horizontal line would indicate theta reset to a fixed phase. Each scatterplot is accompanied by histograms of the phase distributions. The mean resultant length (R) and Rayleigh test P-value are reported for cases where P<0.05. Note that even in cases with a significant stimulus-evoked alignment of theta phase, the actual perturbation of ongoing theta oscillations is very subtle.

The following figure supplements are available for figure 5:

**Figure supplement 1.** Stimulus-evoked changes in hippocampal theta oscillations while sitting.

**Figure supplement 2.** Stimulus-evoked initiation of hippocampal theta oscillations.

**Figure supplement 3.** Methods for theta phase analysis.

**Figure supplement 4.** Phase precession is unaffected by the eyeblink stimuli.

Previous studies have also reported that the CS can reset the phase of hippocampal theta oscillations (*Nokia et al., 2010*; *Darling et al., 2011*). Given the lack of CS-evoked firing rate changes while running, could the CS onset be encoded by this phase reset instead? Analogous to previous studies, we find evidence for an alignment of theta phase to the CS onset. However, a direct comparison of pre- and post-stimulus phase reveals that this effect is due to a slight perturbation of the ongoing theta phase (*Figure 5*). In particular, this analysis rejects the notion of theta reset to a specific phase. Furthermore, this phase perturbation is present in only one out of three animals, indicating that it is not necessary for performing the task. Moreover, we did not find evidence that theta phase precession was affected by the CS onset (*Figure 5—figure supplement 4*). Note, however, that this analysis does not rule out the possibility that the CS may modulate the degree of theta phase-locking across brain areas.

Finally, extensive lesion studies have demonstrated the critical importance of the hippocampus in trace eyeblink conditioning (*Kim et al., 1995*; *Takehara et al., 2003*). How do our observations fit with this literature? We found that CS-evoked changes in dorsal CA1 firing rates and theta phase are

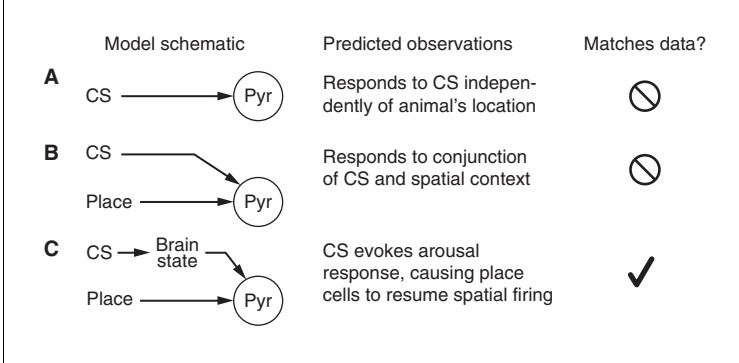

**Figure 6.** Models of CS-responsiveness. (**A**) *CS cells* respond to the CS independently of place. None of the recorded cells consistently responded in space-invariant manner. (**B**) *CS-place cells* encode a context-specific representation of the non-spatial stimulus. The lack of CS responses when the animal is running is not consistent with this model. (**C**) Place cells encode purely spatial information, but arousal-mediated activation of place cells can produce changes in firing following a non-spatial stimulus.

not necessary for task performance—these responses are absent during running, but performance is not impaired—yet dorsally restricted lesions still produce learning deficits (*Takehara et al., 2003*). Several possible explanations may account for this apparent discrepancy: (1) Dorsal hippocampal lesions may disrupt activity in other hippocampal areas (e.g. ventral CA1), which may contain genuinely CS-responsive neurons. (2) The encoding of the CS may be more subtle than changes in firing rate, but still detectable by postsynaptic targets. (3) The critical engagement of the hippocampus may not occur only during the trial itself. Memory consolidation is hippocampally dependent and proceeds without additional training (*Takehara et al., 2003*; *Takehara-Nishiuchi and McNaughton, 2008*), indicating that some eyeblink-related activity occurs well after the stimulus presentation. (4) Even though spatial information may appear irrelevant in this task, the presence of ongoing place-specific firing may still be necessary for learning space-invariant responses. For example, awareness of the context may be necessary for the animal to attend to the appropriate stimuli, or learning a context-dependent association may be a necessary stepping-stone to developing a space-invariant association (*Nadel and Willner, 1980*).

These findings suggest that the dorsal hippocampus transmits exclusively spatial signals, even in a hippocampally dependent non-spatial task, and constrain the ways in which it may contribute to memory processing. These experiments also highlight the importance of accounting for brain state in studies where alertness changes may confound the observed selectivity to experimental stimuli.

## Materials and methods

### Electrophysiological recordings

We recorded from three male Long-Evans rats, 3–5 months old at the time of surgery. We chronically implanted each animal with a differential pair of EMG recording wires in the left eyelid muscle, a pair of bipolar stimulation electrodes in the same area, and a microdrive array that allowed independent adjustment of 24 tetrodes. In two animals (A and B), these tetrodes were targeted to the right dorsal hippocampus; animal C received a bilateral implant targeting the dorsal hippocampus in both hemispheres. We lowered these tetrodes to their targets in the dorsal CA1 cell layer over the course of several days.

Electrophysiological signals were buffered on the headstage, amplified, and digitally recorded for offline analysis. This data acquisition system also received timestamps from the video recording system and the stimulus delivery system to allow these data streams to be synchronized for analysis. All data analysis was performed in MATLAB using the DataJoint data processing chain toolbox (*Yatsenko et al., 2015*).

At the end of the experiment, we marked the final electrode locations using electrolytic lesions. Animals were then euthanized and intracardially perfused with 4% paraformaldehyde. We reconstructed the recording sites using a combination of high-resolution magnetic resonance imaging (*Lubenov and Siapas, 2009*) and histology (brains were cryoprotected, embedded in gelatin, frozen, and sectioned at 20 μm; *Figure 1—figure supplement 1*).

All animal procedures were in accordance with the National Institutes of Health (NIH) guidelines, and with the approval of the Caltech Institutional Animal Care and Use Committee.

## Training environment

All experiments were performed in an acoustically and electromagnetically shielded room containing a sleepbox and a 180 cm long, 7.5 cm wide linear track. Each end of the track terminates in a 22 × 16 cm endbox that contains a water port for reward delivery. An overhead video system tracks LED markers on the tetrode headstage to provide a measurement of animal position.

Animals were housed in the sleepbox within the recording environment. An automated direct-current lighting system provided a 12-hr light cycle with light onset at 9am.

After recovering from surgery, animals were trained to traverse the linear track for water reward. Animals were given at least 12 days to habituate to the environment and the linear track task before starting eyeblink conditioning.

## Conditioning procedure

We used a 250 ms, 5 kHz tone as the CS, a 250 ms stimulus-free trace interval, and 10 ms bipolar stimulation of the eyelid muscle as the US. Lesion studies in rats have shown that this trace interval requires the hippocampus for successful acquisition (*Weiss et al., 1999*). The CS was delivered from a speaker located above the center of the track. The loudness of the tones used during training was determined during a tone-only habituation session by gradually increasing the loudness until startle responses (0–80 ms after CS onset) are detected in the eyelid EMG, then lowering the loudness until these responses disappear. We did not measure the CS loudness along the track, but afterwards verified that conditioned responses could be observed at all locations.

For all animals, we delivered eyeblink conditioning trials at random locations while they were running on the track. Animal C also received trials at random time intervals. Animals received between 38 and 186 eyeblink trials per training session (an average of 109), depending on how much they ran on the track. All animals received two training sessions per day, at the beginning and end of the 12-hr light cycle.

## Behavioral analysis

After filtering the differential EMG data with a 120–960 Hz passband, we computed its root-mean-square (RMS) value in three 100 ms windows: 'baseline' from −100 to 0 ms relative to CS onset, 'CR' from 380 to 480 ms (US onset occurs at 500 ms), and 'control' from −580 to −480 ms. Trials in which [CR − baseline] exceeded the 95th percentile of that animal's [control − baseline] were considered blinks.

## Spike detection and sorting

After filtering the tetrode data with a 600–6000 Hz passband, we detected and aligned spikes on peaks in the nonlinear energy operator (*Mukhopadhyay and Ray, 1998*). Spikes were clustered by fitting a mixture model in a 12-dimensional feature space (three principal components per tetrode channel) (*Calabrese and Paninski, 2011*; *Ecker et al., 2014*). A cluster was considered a single unit if it was well-isolated (sum of false positive and negative rates < 10%), had a refractory period > 1.5 ms, had a spike amplitude > 120 μV, and its spike waveforms remained stable throughout the course of the recording.

These criteria identified 1585 single units, of which 1264 we deemed to be putative pyramidal cells based on firing rate (<4 Hz), spike width (>0.25 ms), and bursting (>5% of inter-spike intervals are <10 ms). Each of these 1264 units is treated independently in our analysis, even though several are likely to be repeated observations of the same neuron. We estimate that we recorded only 400–500 unique pyramidal cells.

## Place field analysis

We constructed place field maps by dividing the number of spikes in each location by the occupancy of that location (*Rich et al., 2014*). Track rate maps are one-dimensional, computed for each direction independently, and smoothed with a σ = 3 cm Gaussian kernel. Endbox rate maps are two-dimensional, non-directional, and smoothed with σ = 1 cm.

Only periods where the animal was moving (velocity > 2 cm/s) were included in the construction of the firing rate maps. Furthermore, the 2-s window following the CS onset was excluded so that eyeblink responses would not influence the construction of the place field maps. Areas with an occupancy < 50 ms/cm$^2$ were considered to have an indeterminate spatial firing rate, and any eyeblink trials that occurred in these locations were excluded from analysis.

We found that 38% of cells were not active in this environment (i.e. did not have a place field with peak > 2 Hz), but this underestimates the actual fraction of silent cells, as we are unable to reliably cluster single units with an overall firing rate < 0.01 Hz.

The *place field intensity* on a given trial is the value of the place field map at the location in space where the trial was delivered. Since the animal may be moving, this is taken as the average over the animal's trajectory:

$$I_{place} = \frac{1}{t_2 - t_1} \int_{t_1}^{t_2} M(x(t), y(t)) \, dt,$$

where $M(x, y)$ is the place field map for this cell and $x, y$ are the animal's trajectory over time. $t_1$ and $t_2$ define the time window for averaging. When sorting rasters (*Figures 2B–E*, *3A–B,* and *4*), this corresponds to the window visible in the figure; for analysis of CS response (*Figures 2F* and *3C*), this corresponds to the 500 ms post-CS analysis window.

## Statistical analysis of CS response

All analysis of CS response is based on 500 ms windows immediately prior to ('pre-CS') and immediately following ('post-CS') the CS onset.

To characterize a cell's CS response, we used the two-sided Wilcoxon signed-rank test without correction for multiple comparisons. We found 123 cells (10% of all pyramidal cells) that were significant at a p<0.01 level. Of these, 87 increased their firing rate and 36 decreased their firing rate. However, this statistical test should be interpreted as a heuristic measure only, as the influences of spatial tuning and brain state violate the test's assumption that samples are independent and identically distributed.

To quantify the reliability of a cell's CS response (*Figure 1F*), we looked for differences in pre- vs. post-CS firing on a trial-by-trial basis. For a rate-increasing cell like the one shown in *Figure 1E*, the observed response reliability is the fraction of trials in which it fired more spikes post-CS than pre-CS. The expected response reliability is the probability of this occurring given the cell's average pre- and post-CS firing rates, assuming that spike counts are Poisson distributed. Specifically,

$$P_{k_2 > k_1} = \sum_{k_1 = 0}^{\infty} \sum_{k_2 = k_1 + 1}^{\infty} f(k_1; \lambda_1) f(k_2; \lambda_2),$$

where $f(k; \lambda)$ is the Poisson probability mass function and $\lambda_1, \lambda_2$ are this cell's average number of pre- and post-CS spikes, respectively.

To quantify the spatial tuning of the CS-evoked firing (*Figure 2F*), we computed Spearman's rank correlation coefficient between the CS-evoked firing and the place field intensity at the trial delivery locations. Cells that never fired any spikes following the CS were assigned a correlation of zero.

## Ripple analysis

We detected ripples by looking for high-frequency oscillations in the local field potential (LFP). First, we filtered the tetrode data with an 80–250 Hz passband and looked for peaks in the RMS power. For each detected peak, we determined its frequency using a short-time Fourier transform (STFT) centered around the event and considered it a putative ripple if its frequency was between 120 and 250 Hz and it was detected on at least 3 tetrodes.

## Theta analysis

We analyzed the LFP to identify the effects of the CS/US on the hippocampal theta rhythm. For each animal, we used a single tetrode for all theta analysis. We chose tetrodes superficial to the cell layer (as determined by sharp wave polarity) so that small changes in the tetrode depth would have a minimal effect on the recorded theta phase (*Lubenov and Siapas, 2009*).

For each trial, we estimate the instantaneous theta frequency and phase using a short-time Fourier transform (STFT) with a 400 ms Kaiser window (α = 1). At each time point, we determine the instantaneous theta frequency by looking for a peak in the spectral power density in the 4–12 Hz range. We then obtain the instantaneous theta amplitude and phase from the modulus and argument of the STFT at that frequency (*Figure 5—figure supplement 3B*). The mean resultant length (*Figure 5D*) is given by

$$R(t) = \left| \frac{1}{N} \sum_{n=1}^{N} e^{i\phi_n(t)} \right|,$$

where $\phi_n(t)$ is the theta phase for trial $n$ at time $t$. This measure of circular concentration ranges between 0 (phases are symmetrically distributed, e.g. uniform) and 1 (phases are all concentrated at a single value).

This STFT-based approach guarantees that our theta estimates cannot be influenced by any features in the data more than ± 200 ms away. For example, we often observed a high-amplitude transient following the US delivery. This clearly skews the estimated theta phase in the vicinity of the US, but our analysis approach ensures that phase estimates 200 ms before or after the transient are unaffected. In contrast, techniques that involve the Hilbert transform or an analysis of IIR-filtered signals cannot guarantee that that the influence of such transients remain localized to a fixed time interval.

These guarantees ensure that our phase continuity analysis (*Figure 5E*) does not introduce spurious autocorrelations. The pre-CS window ends at the CS onset, and the post-CS window starts 100 ms after that (*Figure 5—figure supplement 3C*), therefore the pre- and post-CS phase estimates are derived from independent segments of data. To assist in comparison, we use the corresponding frequency estimates to extrapolate both phase estimates to the same point in time (the midpoint of this 100 ms gap, i.e. 50 ms after the CS onset).

## Acknowledgements

We thank C Wierzynski for development of the experimental paradigm with tetrode recordings, A Hoenselaar for development of the data acquisition system and data processing toolchain, J Mok for assistance with histology, and S Cassenaer, B Hulse, and B Sauerbrei for valuable comments on the manuscript.

## Additional information

### Funding

| Funder | Grant reference number | Author |
| --- | --- | --- |
| Larry L. Hillblom Foundation | | Maria Papadopoulou |
| National Science Foundation | IOS-1146871 | Athanassios G Siapas |
| National Institutes of Health | 1DP1OD008255 | Athanassios G Siapas |
| G Harold and Leila Y. Mathers Foundation | | Athanassios G Siapas |
| Gordon and Betty Moore Foundation | | Athanassios G Siapas |
| National Institutes of Health | 5DP1MH099907 | Athanassios G Siapas |

The funders had no role in study design, data collection and interpretation, or the decision to submit the work for publication.

## Author contributions

KQS, Acquisition of data, Analysis and interpretation of data, Drafting or revising the article; EVL, Conception and design, Acquisition of data, Analysis and interpretation of data, Drafting or revising the article; MP, Acquisition of data, Drafting or revising the article; AGS, Conception and design, Analysis and interpretation of data, Drafting or revising the article

## Author ORCIDs

Kevin Q Shan, http://orcid.org/0000-0002-2621-1274
Evgueniy V Lubenov, http://orcid.org/0000-0002-1099-944X
Athanassios G Siapas, http://orcid.org/0000-0001-8837-678X

## Ethics

Animal experimentation: This study was performed in strict accordance with the recommendations in the Guide for the Care and Use of Laboratory Animals of the National Institutes of Health. All of the animals were handled according to approved institutional animal care and use committee (IACUC) protocols (#1465) of the California Institute of Technology. All surgeries were performed under aseptic conditions and under isoflurane anesthesia.

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
