## [Decision Letter]

Thank you for submitting your article "Spatial tuning and brain state account for dorsal hippocampal CA1 activity in a non-spatial learning task" for consideration by *eLife*. Your article has been reviewed by two peer reviewers, and the evaluation has been overseen by a Reviewing Editor and Eve Marder as the Senior Editor.

The reviewers have discussed the reviews with one another and the Reviewing Editor has drafted this decision to help you prepare a revised submission.

Summary:

This is a very interesting study which identifies a novel spatial correlate of the hippocampal substrate of trace eyeblink conditioning. The authors examined hippocampal pyramidal cell firing during eyeblink conditioning and investigated the relationship between this and spatial firing. They report that conditioned eyeblink unit responses were not consistent, occurring preferentially within the independently mapped place field. Outside of the place field the CS silenced the cell. Furthermore, CS-evoked responses did not occur during running but only when animal was inactive. What seems to be happening is that when the animal becomes inactive in the place field the firing rate drops below what would be predicted on the basis of the active field rate and this active-movement rate is restored by the CS. As has been shown before, sharpwave ripples occurred when the animal was not moving and were inhibited by the CS. The authors conclude that the simplest explanation for their results is that all of the pyramidal cells are place cells, and that, following conditioning, the CS triggers an arousal response which leads to the resumption of the spatial firing in the place field. Importantly they find no evidence for an independent response to the CS which is not dependent on the animal's spatial location.

The paper is well-written and well-illustrated and makes an important contribution to our understanding of the role of the hippocampus in conditioned responses. Essentially it shows that these are dependent on the spatial function of this brain region and are merely a reflection of one aspect of this spatial function, namely that as an animal sits quietly in the place field, its arousal level drops with a concomitant decrease in firing of the place cell. The CS increases arousal and via this change in state restores the firing of the cell. There is no evidence for a role of the hippocampus conditioning independent of this arousal influence on spatial firing. Importantly, the results rule out the CS- spatial location conjunction hypothesis since the CS does not fire the cells in the place field during running.

While the primary results are presented clearly, there are gaps in the analysis that should be addressed. In particular, there is a notable absence of analysis of theta activity and its relationship to conditioning, and CS-dependent hippocampal response.

Essential revisions:

1) In the Discussion, the authors show how their results explain all of the previous findings on the role of hippocampal pyramidal cells in conditioning experiments. On the other hand, they admit that they cannot account for the effect of the hippocampal lesions on eyeblink conditioning. Their only suggestion is that spatial signals are critical for spacing-invariant learning, but this does not seem very compelling or plausible. A possibility not considered is that trace eyeblink conditioning is context dependent and that spatial context is dependent on the hippocampus. Nadel and Willner (1980) originally suggested that the role of the hippocampus in some of these conditioning tasks involved conditioning to the spatial context which might provide the arousing or occasion-setting input which promotes the association between the CS and US especially if there is an interval between the two (see also O'Keefe 1999).

2) Since animals spend most of their time in the end goal boxes and therefore most of the CS's were delivered there (see Figure 2) there is a potential confound between location and behavior. It would have been better if the animal had been required to move and remain still at all locations along the track. Such data would be interesting (though optional)? If the authors do not already have this data, this issue should be discussed.

3) The Moira paper is cited but not really adequately discussed despite the clear overlap in approach and findings. Both involve auditory tone CS conditioning of eyelid shock US response (freezing in their case vs. eyeblink in this study) and both demonstrate dependency on existing place field responses. The observation of CS-evoked responses during ripple-associated immobility is novel and was not addressed in the Moira study, but in that study they did examine questions regarding theta synchronization and the possibility of type II theta during CS-associated immobility. Given the conclusion of lack of CS-independent hippocampal firing rate response, these should be addressed in the present study, with further attention to possible CS-induced changes in hippocampal response that may be relevant such as CS-dependent theta-phase effects (theta phase resetting), and impact on phase precession (if possible).

4) While it does not undermine the central finding of CS-evoked responses that are predicted by underlying place fields, the absence of an unpaired CS control condition does weaken the overall interpretation in that it does not afford the opportunity to differentiate potential hippocampal correlates of simple CS-related responses from successful CS-US association. The potential contribution of CS salience and associated arousal is discussed but is not explicitly tested.

5) Figure 1 seems to show CS associated responses restricted to the early trials suggesting perhaps a training effect. This raises the questions as to whether the reported overall place field predicted CS response changes over the course of training. Is there a change in measures such as CS response reliability over training (e.g. comparing trials 1-7 with below 50% performance and trials 8-14 with above 50% performance)?

6) Given the absence of a CS-independent hippocampal response, a related question is whether the conditioning would be successful if CS-US pairings only occurred during running when no CS modulation of hippocampal response is observed? It would seem that this is a strong and surprising prediction which could be tested.

7) Additional information should also be provided on the distribution of place fields and overall behavior. Is hippocampally-dependent conditioning potentially derived from a subset of the CS-US pairings (perhaps those associated with stopping points as suggested in the example in Figure 1)? What is the spatial distribution of stopping locations, and what is the result of restricting the place field and CS response analyses to those locations where CS-associated responses are observed (even if they are arousal state related as the authors suggest)? What is the result of a within-cell comparison of CS response to running and stopping (cells for which CS response can be compared during stopping and running through the same location) as opposed to the across-cell population analysis currently used which may suffer from biases sampling bias due to non-uniform running/stopping behavior as a function of location? The analysis shown in Figure 3 may be adequate to address this concern given a reasonably uniform distribution of stopping locations.

8) It is interesting that from the data shown in Figure 2–Figure 4, the US response also seems to show the same place field dependent modulation, but there is no analysis or discussion of this.

---

## [Author Response]

*The paper is well-written and well-illustrated and makes an important contribution to our understanding of the role of the hippocampus in conditioned responses. Essentially it shows that these are dependent on the spatial function of this brain region and are merely a reflection of one aspect of this spatial function, namely that as an animal sits quietly in the place field, its arousal level drops with a concomitant decrease in firing of the place cell. The CS increases arousal and via this change in state restores the firing of the cell. There is no evidence for a role of the hippocampus conditioning independent of this arousal influence on spatial firing. Importantly, the results rule out the CS- spatial location conjunction hypothesis since the CS does not fire the cells in the place field during running.*

While the primary results are presented clearly, there are gaps in the analysis that should be addressed. In particular, there is a notable absence of analysis of theta activity and its relationship to conditioning, and CS-dependent hippocampal response.

We appreciate the reviewers’ thoughtful and constructive comments, which have helped us improve the manuscript substantially. In response to their feedback, we have added further analysis and discussion that we believe address the issues they have raised. The most important changes include the following:

We analyze hippocampal theta oscillations for perturbations associated with the CS or US. We present this analysis in a set of new figures (Figure 5 and associated supplements) and discuss these results in the text.We address a potential confound between the animal’s behavior and location. We now analyze the CA1 response separately for trials delivered on the track and the endboxes (Figure 3—figure supplement 2), which confirms that it is indeed the behavior (running vs. sitting), rather than location (track vs. endbox), that explains the difference in observed CS response.We present additional single-cell examples and analysis. We believe these additional illustrations (Figure 2—figure supplement 1 through 3) complement the pooled analysis we present in the main figures.

We discuss these revisions in more detail below in a point-by-point response to the reviewers’ comments.

*Essential revisions:*

1) In the Discussion, the authors show how their results explain all of the previous findings on the role of hippocampal pyramidal cells in conditioning experiments. On the other hand, they admit that they cannot account for the effect of the hippocampal lesions on eyeblink conditioning. Their only suggestion is that spatial signals are critical for spacing-invariant learning, but this does not seem very compelling or plausible. A possibility not considered is that trace eyeblink conditioning is context dependent and that spatial context is dependent on the hippocampus. Nadel and Willner (1980) originally suggested that the role of the hippocampus in some of these conditioning tasks involved conditioning to the spatial context which might provide the arousing or occasion-setting input which promotes the association between the CS and US especially if there is an interval between the two (see also O'Keefe 1999).

In our Discussion, we described four potential explanations for the apparent discrepancy between the effect of hippocampal lesions and our finding that CS-evoked changes in dorsal CA1 firing rates are not necessary for task performance: (1) lesions may disrupt other parts of the hippocampus or other brain areas, (2) the encoding of the CS may be more subtle than firing rate changes, (3) the hippocampal influence may not occur during the trial itself, and (4) spatial information may be necessary for this non-spatial task. We did not intend to endorse only one of these explanations, and have reworded our conclusion to avoid this implication.

We have also clarified our suggestion that spatial information may be necessary for space-invariant learning. Providing the occasion-setting input that promotes the learned association—even when the CS and US are non-spatial in nature—is consistent with the role that we envision the place cell firing may play, and we have incorporated the reviewers’ suggestion in our Discussion.

2) Since animals spend most of their time in the end goal boxes and therefore most of the CS's were delivered there (see Figure 2) there is a potential confound between location and behavior. It would have been better if the animal had been required to move and remain still at all locations along the track. Such data would be interesting (though optional)? If the authors do not already have this data, this issue should be discussed.

The distribution of trial locations in Figure 2 is just one example. We have added new figures showing additional examples (Figure 2—figure supplement 2) and the overall distribution of trial delivery locations (Figure 1—figure supplement 2).

To address the potential confound between location and behavior, we show that animals moved and remained still throughout the environment, and that it is the behavior (running vs. sitting) rather than the location (track vs. endbox) that explains the difference in the observed CS response (Figure 3—figure supplement 2).

3) The Moira paper is cited but not really adequately discussed despite the clear overlap in approach and findings. Both involve auditory tone CS conditioning of eyelid shock US response (freezing in their case vs. eyeblink in this study) and both demonstrate dependency on existing place field responses. The observation of CS-evoked responses during ripple-associated immobility is novel and was not addressed in the Moira study, but in that study they did examine questions regarding theta synchronization and the possibility of type II theta during CS-associated immobility. Given the conclusion of lack of CS-independent hippocampal firing rate response, these should be addressed in the present study, with further attention to possible CS-induced changes in hippocampal response that may be relevant such as CS-dependent theta-phase effects (theta phase resetting), and impact on phase precession (if possible).

Figure 2 (“Spatial tuning modulates CA1 responses to eyeblink stimuli”) is indeed consistent with the key finding of Moita et al. (2003) that “CS responses of place cells are gated by the location-specific activity of the cells.”

However, our key finding is that these CS-evoked changes in firing rate are absent when the animal is running. This observation rules out the CS-place conjunctive coding hypothesis endorsed by Moita et al., and we have expanded our discussion of this result.

The key differences in approach that enabled this new finding include a less disruptive conditioned response (blinking vs. freezing) and a shorter CS-US interval (500 ms vs. 20 seconds), which allowed us to study the case where the animal remained running throughout the trial. A larger sample size (1264 vs. 47 cells) enabled us to perform a more quantitative analysis, and studying a hippocampally-dependent task (trace eyeblink conditioning vs. cued fear conditioning) underscores the relevance of the results.

Following the reviewers’ suggestion, we now include analysis of CS- and US-related theta phase effects (Figure 5). Although we find a statistically significant alignment of theta phase following the CS and US, we observed this in only one of three animals and we show that it arises from a subtle perturbation of ongoing theta rhythm rather than a reset to a fixed phase. We also investigated the impact on phase precession and found no evidence for differences between CS-evoked firing and usual place cell activity (Figure 5—figure supplement 4).

4) While it does not undermine the central finding of CS-evoked responses that are predicted by underlying place fields, the absence of an unpaired CS control condition does weaken the overall interpretation in that it does not afford the opportunity to differentiate potential hippocampal correlates of simple CS-related responses from successful CS-US association. The potential contribution of CS salience and associated arousal is discussed but is not explicitly tested.

We agree with the reviewers that the inclusion of an unpaired CS condition would have been a valuable addition to our experimental design. However, the key result in our study is the finding that CS-evoked changes in firing rate are absent when the animal is running (Figure 3), without any impairment of task performance (Figure 3—figure supplement 1). Regardless of whether these firing rate changes are simple CS-related responses or the product of successful CS-US association, we have shown that they are not necessary to perform the task.

Previous studies have shown that pseudoconditioned animals (those that received unpaired CS and US) exhibit fewer CS-evoked changes in firing rate than trained animals (Weiss et al., 1996), and this difference arises around the same time that trained animals begin to develop behavioral responses to the CS (McEchron and Disterhoft, 1997). This is consistent with our interpretation that the unpaired CS does not acquire behavioral relevance and thus has little efficacy in driving a brain state change.

We explored the potential contribution of CS salience and associated arousal by showing that dorsal CA1 firing is agnostic to the CS when the animal is already engaged in an active brain state (i.e. running). Demonstrating the converse—that any salient stimulus would produce the same response as the paired CS-US when presented in the appropriate brain state—would be a more explicit test of this hypothesis. Identifying a set of equivalently salient stimuli and performing a within-cell comparison of responses to the CS and responses to these other stimuli is a promising direction for future studies.

5) Figure 1 seems to show CS associated responses restricted to the early trials suggesting perhaps a training effect. This raises the questions as to whether the reported overall place field predicted CS response changes over the course of training. Is there a change in measures such as CS response reliability over training (e.g. comparing trials 1-7 with below 50% performance and trials 8-14 with above 50% performance)?

Figure 1 shows responses from a single training session (out of 14 total for this animal). We have included additional examples from other cells in the same session (Figure 2—figure supplement 1), which show that a different subset of place cells become active in later trials, when the animal moves to a different location in the environment. These data show that the animal’s location, rather than the course of training, determines whether a given cell fires after the CS.

Nonetheless, we agree with the reviewers that learning-related changes are a very compelling topic for further investigation. However, in light of our finding that the CS response depends on the animal’s brain state (Figure 3 and Figure 4), we do not believe that we can satisfactorily perform this analysis within the scope of this paper. In freely-behaving animals, many behaviors change over the course of training: besides learning the conditioned response, the animals also ran faster and exhibited shorter bouts of immobility. Hence it is likely that the distribution of pre-trial brain state also shifts over the course of training. Given the striking differences in CS response under different brain states, it is necessary to distinguish genuine learning-related changes from the side effects of this shift in pre-trial brain state. Additional controls are necessary to properly interpret any observed changes over the course of training.

6) Given the absence of a CS-independent hippocampal response, a related question is whether the conditioning would be successful if CS-US pairings only occurred during running when no CS modulation of hippocampal response is observed? It would seem that this is a strong and surprising prediction which could be tested.

Indeed we find that conditioning was successful even when an animal received the vast majority of conditioning trials while running (animal A, Figure 1—figure supplement 2).

Although this is only one animal, it suggests that CS-evoked changes in the firing of dorsal CA1 pyramidal cells are not necessary for learning the CS-US association. As we discuss, there are many ways in which the dorsal hippocampus may still contribute to learning and performing the task, but it seems that CS-evoked firing rate changes are not one of them.

*7) Additional information should also be provided on the distribution of place fields and overall behavior.*

We have added more single-cell examples and a discussion of their corresponding place fields (Figure 2—figure supplement 2). This is in addition to the overall distribution of place fields that we previously presented (Figure 2). We have also added figures showing the distribution of animal location and behavior during eyeblink trials (Figure 1—figure supplement 2, Figure 3—figure supplement 2).

*Is hippocampally-dependent conditioning potentially derived from a subset of the CS-US pairings (perhaps those associated with stopping points as suggested in the example in Figure 1)?*

Previous studies have shown that presenting CS-US pairings during periods of high theta power (Berry and Thompson, 1978; Nokia et al., 2009) or ripples (Nokia et al., 2010) are both associated with faster learning, suggesting that a subset of CS-US pairings may be disproportionately responsible for conditioning.

However, identifying the subset of trials responsible for conditioning would require comparing the learning rates of animals trained under different conditions, so in general this is not a question that we are able to answer with our data.

As to the specific hypothesis that hippocampally-dependent conditioning is associated with eyeblink trials delivered while the animal is sitting, we found that conditioning was successful even when an animal received the vast majority of conditioning trials while running (animal A, Figure 1—figure supplement 2). This suggests that sitting still—and the associated CS-evoked changes in CA1 pyramidal cell firing—is not necessary for successful conditioning.

Berry SD, Thompson RF. 1978. Prediction of learning rate from the hippocampal electroencephalogram. *Science***200**:1298–1300. doi:10.1126/science.663612

*What is the spatial distribution of stopping locations, and what is the result of restricting the place field and CS response analyses to those locations where CS-associated responses are observed (even if they are arousal state related as the authors suggest)?*

We have added the spatial distribution of running and sitting locations to Figure 3, and now analyze the running/stopping behavior separately for the track and endboxes (Figure 3—figure supplement 2). We found that CS-associated responses may be observed in all locations throughout the environment, and that it is indeed the behavior (running vs. sitting) rather than the location (track vs. endbox) that explains the difference in the observed CS response.

What is the result of a within-cell comparison of CS response to running and stopping (cells for which CS response can be compared during stopping and running through the same location) as opposed to the across-cell population analysis currently used which may suffer from biases sampling bias due to non-uniform running/stopping behavior as a function of location? The analysis shown in Figure 3 may be adequate to address this concern given a reasonably uniform distribution of stopping locations.

Given the size of the environment, any single training session contains too few trials that satisfy the desired criteria (i.e. stopping and running through the same location) to make a reliable within-cell comparison.

To address the non-uniform running/stopping behavior as a function of location, we analyzed the track and the endboxes separately (Figure 3—figure supplement 2). This shows that it is the behavior (running vs. sitting) rather than the location (track vs. endbox) that explains the difference in the observed CS response.

Furthermore, we have added a new figure showing, on a cell-by-cell basis, that the observed CS- and US-evoked firing aligns with the cell’s place field intensity at the trial delivery location (Figure 2—figure supplement 3).

*8) It is interesting that from the data shown in Figure 2–Figure 4, the US response also seems to show the same place field dependent modulation, but there is no analysis or discussion of this.*

We now show the same level of analysis for the US that we do for the CS across these figures.

We have chosen to focus our discussion on the CS response because it seems more pertinent to learning and performing the task, and because the US response follows essentially the same pattern. Furthermore, analysis of the US response is potentially confounded by reactive behaviors, such as sudden head movements, that are associated with the unconditioned response.